# Composition of Vapor–Liquid–Solid III–V Ternary Nanowires Based on Group-III Intermix

**DOI:** 10.3390/nano13182532

**Published:** 2023-09-11

**Authors:** Vladimir G. Dubrovskii

**Affiliations:** Faculty of Physics, St. Petersburg State University, Universitetskaya Emb. 13B, St. Petersburg 199034, Russia; dubrovskii@mail.ioffe.ru

**Keywords:** III–V nanowires, vapor–liquid–solid growth, group-III intermix, vapor–solid distribution, compositional control

## Abstract

Compositional control in III–V ternary nanowires grown by the vapor–liquid–solid method is essential for bandgap engineering and the design of functional nanowire nano-heterostructures. Herein, we present rather general theoretical considerations and derive explicit forms of the stationary vapor–solid and liquid–solid distributions of vapor–liquid–solid III–V ternary nanowires based on group-III intermix. It is shown that the vapor–solid distribution of such nanowires is kinetically controlled, while the liquid–solid distribution is in equilibrium or nucleation-limited. For a more technologically important vapor-solid distribution connecting nanowire composition with vapor composition, the kinetic suppression of miscibility gaps at a growth temperature is possible, while miscibility gaps (and generally strong non-linearity of the compositional curves) always remain in the equilibrium liquid–solid distribution. We analyze the available experimental data on the compositions of the vapor–liquid–solid Al_x_Ga_1−x_As, In_x_Ga_1−x_As, In_x_Ga_1−x_P, and In_x_Ga_1−x_N nanowires, which are very well described within the model. Overall, the developed approach circumvents uncertainty in choosing the relevant compositional model (close-to-equilibrium or kinetic), eliminates unknown parameters in the vapor–solid distribution of vapor–liquid–solid nanowires based on group-III intermix, and should be useful for the precise compositional tuning of such nanowires.

## 1. Introduction

It is known that III–V ternary nanomaterials with widely tunable compositions are paramount for the fundamental research of semiconductor properties at the nanoscale, bandgap engineering, and fabrication of functional heterostructures for different device applications in nanoelectronics and optoelectronics [1]. Furthermore, III–V ternary nanowires (NWs) and heterostructures within such NWs are an emerging class of nanomaterials which provides almost unlimited opportunities for bottom-up bandgap design [2,3]. Due to a very efficient relaxation of elastic stress on NW sidewalls, III–V ternary NWs and III–V NW heterostructures are much less restricted by lattice mismatch [4] and can be grown on dissimilar Si substrates without forming dislocations [5]. These properties cannot be achieved in epi-layers and even in Stranski–Krastanow quantum dots [6]. High-quality III–V ternary NWs on Si substrates are therefore promising for monolithic integration of III–V-based optoelectronics with a Si electronic platform [7]. Most III–V NWs are grown by the vapor–liquid–solid (VLS) method with a metal catalyst droplet [8], which can be either Au or a group-III metal [9], with a self-catalyzed approach.

Compositions of VLS III–V ternary NWs based on group-III intermix, including In_x_Ga_1−x_As [10,11,12,13,14,15,16], In_x_Ga_1−x_P [17], Al_x_Ga_1−x_As [18,19,20,21,22,23], and In_x_Ga_1−x_N [24] material systems, have been extensively studied with different epitaxy techniques versus technologically controlled growth conditions such as temperature and material fluxes. Understanding the growth of III–V ternary NWs and controlling their compositions by growth parameter tuning necessarily requires advanced modeling. Theoretical approaches developed so far (see Refs. [25,26,27] for a review) treat liquid–solid distributions connecting the composition of a VLS ternary A_x_B_1−x_C, pseudo-binary (AC)_x_(BC)_1−x_ NW x to the content of A atoms in liquid y [20,28,29,30,31,32,33,34] or vapor–solid distributions connecting x to the content of A atoms in vapor z [14,21,24,35,36]. According to the general treatment given in Refs. [36,37], the liquid–solid distribution $x\left(y\right)$ of a VLS A_x_B_1−x_C ternary NW based on group-III intermix (with A and B belonging to group III and C belonging to group V) is close to equilibrium [20,28], which is equivalent to the nucleation-limited distribution (with a composition-independent edge energy of a critical two-dimensional (2D) island [29]) derived earlier in Refs. [29,30,31]. This fundamental property is related to the C-poor conditions [36,37,38] for the liquid–solid growth of 2D islands, which is guaranteed for VLS NWs due to an extremely low (~1% or even less) concentration of highly volatile group V atoms in the catalyst droplets. Importantly, a close-to-equilibrium growth regime is independent of the supersaturation level and may occur even under infinitely high supersaturation, simply due to an excess of group-III atoms and a lack of group-V atoms available for growth [36,37]. On the other hand, the vapor–solid distribution $x\left(z\right)$ of III–V ternary NWs based on group-III intermix has been obtained in a kinetic form in Ref. [35], using the assumption of group-V-rich vapor–solid growth and without introducing a liquid droplet directly. The kinetic vapor–solid distribution fits quite well the compositional data of Au-catalyzed InGaAs [11] and InGaP [17] NWs. The first attempt to join liquid–solid and vapor–solid distributions of VLS III–V ternary NWs was made in Ref. [37]. Herein, we further develop this approach by considering the limiting steps of the whole VLS growth process, which appear to be different for group-III and group-V atoms. As a result, we present the equilibrium liquid–solid and kinetic vapor–solid distributions of III–V ternary NWs based on group-III intermix. The obtained analytic form of the vapor–solid distribution contains no parameters of the liquid phase, some of which remain generally unknown, and is very useful for the compositional control over VLS III–V ternary NWs. We consider the available experimental data on the compositions of the VLS Al_x_Ga_1−x_As, In_x_Ga_1−x_As, In_x_Ga_1−x_P, and In_x_Ga_1−x_N nanowires, which are very well described within the model. In particular, we demonstrate that the miscibility gaps in the material systems with strong interactions between dissimilar III–V pairs in solids, such as highly mismatched InGaAs, InGaP, and InGaN ternaries, can be fully circumvented by fast VLS growth kinetics.

## 2. Limiting Steps of Group-III and Group-V Element Incorporation

The VLS growth of a ternary A_x_B_1−x_C NW based on group-III intermix from atomic vapor fluxes $I_{A}$, $I_{B}$, and $I_{C}\;$ is illustrated in Figure 1. The vapor flux of group-V atoms is usually higher or even much higher than the total vapor flux of group-III atoms in Au-catalyzed VLS growth [11,14,15,21]. The total arrival rates of group-III atoms into the droplet shown in the figure are usually enhanced by surface diffusion of A and B adatoms from the NW sidewalls [25,27,35,36,37]. Even with this contribution included, the effective arrival rate of A and B atoms into the droplet cannot be much higher than $I_{C}$. Hence, the vapor phase is group-V rich in most cases. The situation is reversed in a catalyst droplet regardless of whether the VLS growth is Au-catalyzed or self-catalyzed. The group-V concentration in the droplet $\chi_{C}$ is in the order of 0.01 or even less, which is much lower than the total group-III concentration $\chi_{A}+\chi_{B}$ according to the data of Refs. [20,25,26,27,28,29,30,31,32,33,34,35,36,37]. This fundamental property holds regardless of the fact that the droplets catalyzing the VLS growth of InGaAs [15,29,31,32], InGaP [17], or InGaN [24] NWs are In-rich, while the droplets catalyzing the VLS growth of AlGaAs NWs are Ga-rich [20,29,31]. Hence, the liquid–solid growth of a 2D island, or partial NW monolayer, is always group-III rich and is limited by the incorporation of group-V atoms [37].

The condition ${\chi_{C}\ll \chi }_{A}+\chi_{B}$ in liquid at $I_{C}\sim I_{A}+I_{B}$ or even $I_{C}\gg I_{A}+I_{B}$ in vapor can only be satisfied if most group-V atoms desorb from the droplet surface without entering the droplet. A small fraction of group-V atoms that have entered a droplet and diffused to the boundary of a growing 2D island will be subsequently incorporated into an NW with almost 100% probability. Therefore, the liquid–solid interface does not limit the liquid–solid incorporation of these atoms. Rather, the group-V incorporation into a VLS NW is limited by the vapor–liquid interface. The situation is reversed for group-III atoms, which easily enter the droplet through the vapor–solid interface or triple phase line, while their incorporation at the liquid–solid interface is difficult due to a lack of group-V atoms in the droplet. Therefore, the incorporation of group-III atoms into a VLS NW is limited by the liquid–solid interface. The two limiting steps of the VLS growth were discussed a long time ago [39] but not for III–V NWs, where they appear different for group-III and group-V atoms in the same growth process. The impact of this property on the composition of VLS ternary NWs based on group-III intermix will be discussed in detail in this work.

## 3. Liquid–Solid Distribution

The liquid–solid distribution of a VLS III–V A_x_B_1−x_C NW presents the solid composition x as a function of liquid composition
(1)$$y=\frac{\chi_{A}}{\chi_{A}+\chi_{B}},$$
which is given by the ratio of the atomic concentration of species *A* over the total concentration of *A* and *B* atoms in liquid, and is considered as a governing parameter for NW composition. As discussed in detail in Ref. [36] and Ref. [37] specifically for VLS III–V NWs, group-V-poor (or C-poor) conditions for the liquid–solid growth of a partial NW monolayer make the liquid–solid distribution of VLS NWs based on group-III intermix close to equilibrium regardless of the supersaturation level in the droplet with respect to a solid. The equilibrium vapor–solid distribution is obtained from the two conditions of equilibrium in a pseudo-binary system [20,27,28]
(2)$$\mu_{A}+\mu_{C}=\mu_{AC},\;\mu_{B}+\mu_{C}=\mu_{BC}.$$

Here, $\mu_{A}$ and $\mu_{B}$ are the chemical potentials of A and B atoms in liquid in thermal units,
(3)$$\mu_{A}=\mu_{A}^{0}+ln\chi_{A}+\psi_{A},\;\mu_{B}=\mu_{B}^{0}+ln\chi_{B}+\psi_{B},$$
which contain the chemical potentials of pure liquids $\mu_{A}^{0}$ and $\mu_{B}^{0}$, logarithmic terms in concentrations and interaction terms $\psi_{A}$ and $\psi_{B}$. These interaction terms generally depend on the atomic concentrations $\chi_{A}$, $\chi_{B}$, and $\chi_{C}$ (with the Au concentration given by $\chi_{Au}=1-\chi_{A}-\chi_{B}-\chi_{C}$) [25,26,27,28,29,30,31,32,33,34,35,36,37]. The number of variables is reduced from three to two for self-catalyzed VLS NWs with $\chi_{Au}=0$. Within the regular solution model, the chemical potentials of AC and BC pairs in a solid are given by [25,26,27,28,29,30,31,32,33,34,35,36,37]
(4)$$\mu_{AC}=\mu_{AC}^{0}+lnx+\omega {\left(1-x\right)}^{2},\;\mu_{BC}=\mu_{BC}^{0}+\ln(1-x)+\omega x^{2},$$
where $\mu_{AC}^{0}$ and $\mu_{BC}^{0}$ are the chemical potentials of pure binaries AC and BC and ω is the pseudo-binary interaction parameter in thermal units (which may be x-dependent). The temperature-dependent ω values are well-tabulated for each III–V ternary material [26,27]. The pseudo-binary interaction parameter is solely responsible for the miscibility gaps, which appear whenever ω>2 and close to the critical temperature corresponding to ω=2. Using Equations (1)–(4), it is easy to obtain the equilibrium liquid–solid distribution of VLS ternary NWs based on group-III intermix in the form [28]
(5)$$y=\frac{1}{1+f_{l}(x)},\;f_{l}\left(x\right)=\beta_{l}\frac{\left(1-x\right)}{x}e^{\omega (2x-1)},\;\beta_{l}=e^{\Delta \mu_{AC}^{0}-\Delta \mu_{BC}^{0}+\psi_{A}-\psi_{B}}.$$

Here, $\Delta \mu_{AC}^{0}-\Delta \mu_{BC}^{0}=\mu_{A}^{0}+\mu_{C}^{0}-\mu_{AC}^{0}-\left(\mu_{B}^{0}+\mu_{C}^{0}-\mu_{BC}^{0}\right)=\mu_{A}^{0}-\mu_{AC}^{0}-\left(\mu_{B}^{0}-\mu_{BC}^{0}\right)$ is the chemical potential difference, which is related to the affinity of A atoms with respect to B atoms.

As discussed in detail in Refs. [27,28,37], the equilibrium distribution given by Equation (5) is identical to the nucleation-limited distribution of Refs. [30,31], which treats the composition of a critical 2D island at nucleation under the assumption of an x-independent edge energy of the island. The equilibrium liquid–solid distribution requires that the chemical potential differences of AC and BC pairs in liquid and solid equal zero according to Equation (1), while the nucleation-limited distribution of Ref. [30] requires that their difference is zero. The final result for the liquid–solid distribution is, however, identical in both models, because the chemical potential of group-V atoms cancels in Equation (5). This property is very important because it circumvents the uncertainty in the concentration of group-V atoms in droplet $\chi_{C}$, which is beyond the detection limit, and consequently chemical potential $\mu_{C}$ is principally unknown [27,37].

## 4. Vapor–Solid Distribution

One difficulty in joining the liquid–solid and vapor–solid distributions of VLS III–V ternary NWs is that the liquid–solid distributions are obtained by treating the nucleation step [29,30,31] or growth kinetics [32,33,34] of a 2D island, while the average vapor–solid NW growth rate (which can easily be measured in any epitaxy technique) is not determined by the fast step of island growth but rather by the waiting time between the successive nucleation events [40]. However, one can use the general expressions [37]
(6)$$\frac{{\pi R}^{2}}{\Omega_{s}}x\frac{dL}{dt}=V_{A}^{+}-V_{A}^{-},\;\frac{{\pi R}^{2}}{\Omega_{s}}(1-x)\frac{dL}{dt}=V_{B}^{+}-V_{B}^{-},$$
where R is the radius of the NW top, $\Omega_{s}$ is the elementary volume for a III–V pair in a solid, dL/dt is the average axial NW growth rate, $V_{A}^{+}$ and $V_{B}^{+}$ are the total arrival rates of A and B atoms into the droplet, and $V_{A}^{-}$ and $V_{B}^{-}$ are the total rates of A and B atoms leaving the droplet. These equations ensure a time-independent volume of the droplet under steady-state conditions and state that fluxes $V_{A}^{+}-V_{A}^{-}$ and $V_{B}^{+}-V_{B}^{-}$ contribute to the total NW growth rate dL/dt with weights x and 1−x, respectively [20,29,37]. Using Equation (6), we obtain
(7)$$\frac{1-x}{x}=\frac{V_{B}^{+}-V_{B}^{-}}{V_{A}^{+}-V_{A}^{-}}.$$

The arrival rates of group-III atoms are given by
(8)$$V_{A}^{+}=(\sigma_{A}{\pi R}^{2}+\eta_{A}2\pi R\lambda_{A})I_{A},\;V_{B}^{+}=(\sigma_{B}{\pi R}^{2}+\eta_{B}2\pi R\lambda_{B})I_{B}.$$

Here,${\;\lambda }_{A}$ and $\lambda_{B}$ are the effective diffusion lengths of A and B adatoms on the NW sidewalls, $\sigma_{A}$ and $\sigma_{B}$ are the effective adsorption coefficients on the droplet surface accounting for the droplet geometry, and $\eta_{A}$ and $\eta_{B}$ are the effective adsorption coefficients at the NW sidewalls. They include the directions of beams with respect to the droplet/NW surfaces in molecular beam epitaxy (MBE) and the temperature-dependent precursor decomposition efficiencies at these surfaces in metal-organic vapor-phase epitaxy (MOVPE) or hydride vapor-phase epitaxy (HVPE) [37,40,41]. More complex diffusion processes, including surface diffusion from the substrate surface, can be introduced by considering the radius-dependent diffusion lengths $\lambda_{A}$ and $\lambda_{B}\;$[37,40,41,42]. These considerations are identical to those in Ref. [37].

The outgoing fluxes of group-III atoms A and B depend on the chemical potentials of these atoms in liquid, $\mu_{A}$ and $\mu_{B}$ [37,40]. In order to find these in a self-consistent manner, we use the above considerations of the limiting steps of group-III and group-V atom incorporation into a VLS NW. Whenever most of the group-V atoms desorb from the droplet surface, one can assume vapor–liquid equilibrium for group-V atoms at the vapor–liquid interface. Assuming that vapor is a mixture of perfect gases, this equilibrium is given by [43]
(9)$$2\mu_{C}=\mu_{C_{2}}^{g}=\mu_{C_{2}}^{g,0}+ln\left(\frac{I_{C}}{I_{C}^{0}}\right).$$

Here, $\mu_{C_{2}}^{g,0}$ is the chemical potential of pure vapor of *C*_2_ dimers at a growth temperature T and a total vapor pressure $P_{C_{2}}^{0}$, yielding the atomic flux $I_{C}^{0}$. This condition was not used in Ref. [37].

Assuming that group-III atoms do not desorb directly from the droplet surface but rather leave the droplet by “negative” diffusion onto the NW sidewalls and then desorb from there [44,45], the outgoing fluxes of A and B atoms are obtained using the approach of Ref. [37]. We solve steady-state diffusion equations for the adatom concentrations on the NW sidewalls
(10)$$D_{k}\frac{d^{2}n_{k}}{d\xi^{2}}+\eta_{k}I_{k}-\frac{n_{k}}{\tau_{k}}=0,\;k=A,B,$$
with the boundary conditions
(11)$$n_{k}\left(\xi \to \infty \right)=\eta_{k}\tau_{k}I_{k},\;k=A,B.$$
(12)$$n_{k}\left(\xi =0\right)=\frac{1}{\Omega_{l}^{2/3}}e^{\mu_{k}-\mu_{k}^{0}},\;k=A,B.$$

Here,$\;D_{k}$ are the surface diffusion coefficients of A and B adatoms; $\tau_{k}$ are their desorption-limited lifetimes on the NW sidewalls, with the corresponding diffusion lengths $\lambda_{k}=\sqrt{D_{k}\tau_{k}}$ for k= A and B; and ξ is the coordinate along the NW growth axis. Equation (12) requires that adatoms A and B are at equilibrium with liquid at the triple phase line surrounding the liquid–solid interface under the droplet [35,37]. Using Equations (2), (4) and (9), we find
(13)$$e^{\mu_{A}-\mu_{A}^{0}}={\left(\frac{I_{C}}{I_{C}^{0}}\right)}^{1/2}e^{\mu_{AC}^{0}-\mu_{A}^{0}-\mu_{C_{2}}^{g,0}/2}xe^{\omega {\left(1-x\right)}^{2}},\;e^{\mu_{B}-\mu_{B}^{0}}={\left(\frac{I_{C}}{I_{C}^{0}}\right)}^{\frac{1}{2}}e^{\mu_{BC}^{0}-\mu_{B}^{0}-\mu_{C_{2}}^{g,0}/2}(1-x)e^{\omega x^{2}}.$$

Calculating the adatom diffusion fluxes
(14)$$V_{k,diff}=2\pi RD_{k}{\left.\frac{dn_{k}}{d\xi }\right|}_{\xi =0},$$
and separating out the incoming fluxes given by Equation (8), we obtain
(15)$$V_{A}^{-}=2\pi R\frac{\lambda_{A}}{\tau_{A}}\frac{1}{\Omega_{l}^{2/3}}{\left(\frac{I_{C}}{I_{C}^{0}}\right)}^{1/2}e^{\mu_{AC}^{0}-\mu_{A}^{0}-\mu_{C_{2}}^{g,0}/2}xe^{\omega {\left(1-x\right)}^{2}},\;V_{B}^{-}=2\pi R\frac{\lambda_{B}}{\tau_{B}}\frac{1}{\Omega_{l}^{2/3}}{\left(\frac{I_{C}}{I_{C}^{0}}\right)}^{1/2}e^{\mu_{BC}^{0}-\mu_{B}^{0}-\mu_{C_{2}}^{g,0}/2}(1-x)e^{\omega x^{2}}.$$

Using Equations (8) and (15) in Equation (7), and the definition for the fraction of A atoms in vapor,
(16)$$z=\frac{I_{A}}{I_{A}+I_{B}},$$
the vapor–solid distribution of VLS III–V ternary NWs based on group-III intermix is obtained in the form
(17)$$z=\frac{x}{c+\left(1-c\right)x}\left[1+\Gamma (1-x)\left(\frac{\lambda_{A}}{\lambda_{B}}e^{\omega {\left(1-x\right)}^{2}}-b_{g}e^{\omega x^{2}}\right)\right],\;c=\frac{\lambda_{A}}{\lambda_{B}}\left(\frac{\eta_{A}+\sigma_{A}R/2\lambda_{A}}{\eta_{B}+\sigma_{B}R/2\lambda_{B}}\right),\;\Gamma =\frac{1}{\eta_{B}+\sigma_{B}R/2\lambda_{B}}\frac{1}{\tau_{A}\Omega_{l}^{2/3}}\frac{{\left(I_{C}^{0}\right)}^{1/2}}{(I_{A}+I_{B})I_{C}^{1/2}}e^{\mu_{AC}^{0}-\mu_{A}^{0}-\mu_{C_{2}}^{g,0}/2},\;b_{g}=\frac{\tau_{A}}{\tau_{B}}e^{\Delta \mu_{AC}^{0}-\Delta \mu_{BC}^{0}},$$
which is the main result of this work. This distribution has the same form as the kinetic vapor–solid distribution of Ref. [35], which was obtained for the diffusion-induced growth of III–V ternary materials under the assumption of group-V-rich conditions, and without any droplets. The coefficients in Equation (16) are slightly modified with respect to Ref. [35] to account for the VLS growth geometry and desorption of group-III atoms via negative diffusion onto the NW sidewalls. It is interesting to note that the kinetic liquid–solid distribution for VLS III–V ternary NWs, considered in Refs. [24,32,33,34], is similar to Equation (16), with the modified coefficients related to the liquid–solid growth of a ternary island. As discussed above and in more detail in Ref. [37], the VLS growth of ternary NWs based on group–III intermix is not controlled by the liquid–solid incorporation of group–III atoms, and hence the kinetic liquid–solid distribution should not be used for such NWs. However, the vapor–solid distribution has the kinetic form provided that most group–V atoms desorb from the droplet surface, as given by Equation (9) in our model. Equation (17) contains no characteristics of liquid droplets on the NW top, except for the “geometrical” coefficients $\eta_{A}$ and $\eta_{B}$ entering parameter c. Indeed, the diffusion lengths $\lambda_{A}$ and $\lambda_{B}$ describe the diffusion transport of group-III adatoms on the NW sidewalls, the characteristic lifetimes $\tau_{A}$ and $\tau_{B}$ correspond to desorption from the NW sidewalls, whereas the chemical potential $\mu_{C_{2}}^{g,0}$ entering parameter $b_{g}$ is related to the vapor phase. Therefore, our vapor–solid distribution is independent of the liquid state in the droplets and is entirely determined by the vapor fluxes, characteristics of the vapor–solid interface, and the pseudo-binary interaction parameter in solids.

## 5. Equilibrium Vapor–Solid Distribution

The equilibrium vapor–solid distribution, which sets a limit for the solid composition under no-growth conditions, is obtained from
(18)$$V_{A}^{+}=V_{A}^{-},\;V_{B}^{+}=V_{B}^{-},$$
which corresponds to zero NW growth rate according to Equation (6). Using Equation (8) for $V_{k}^{+}$ and Equation (15) for $V_{k}^{-}$, it is easy to obtain
(19)$$z=\frac{1}{1+f_{g}(x)},\;f_{g}\left(x\right)=\beta_{g}\frac{\left(1-x\right)}{x}e^{\omega (2x-1)},\;\beta_{g}=\left(\frac{\eta_{A}+\sigma_{A}R/2\lambda_{A}}{\eta_{B}+\sigma_{B}R/2\lambda_{B}}\right)b_{g}.$$

This equilibrium vapor–solid distribution has the same form as the equilibrium liquid–solid distribution given by Equation (5), but with modified parameters which correspond to the vapor rather than the liquid phase.

## 6. Results and Discussion

According to Equations (5) and (19), the equilibrium liquid–solid and vapor–solid distributions of VLS III–V ternary NWs based on group-III intermix are determined by the two control parameters, the pseudo-binary interaction parameter ω, and the affinity parameter $\beta_{l}$ (for liquid) or $\beta_{g}$ (for vapor). Both equilibrium curves contain a miscibility gap region at ω>2. The affinity parameters $\beta_{l}$ and $\beta_{g}\;$ depend primarily on the exponential term $\exp(\Delta \mu_{AC}^{0}-\Delta \mu_{BC}^{0})$, which is very small for In_x_Ga_1−x_As [29,30,31,35], In_x_Ga_1−x_P [31,35], and In_x_Ga_1−x_N [24] material systems and very large for the Al_x_Ga_1−x_As material system [20,30,31] for any reasonable growth temperatures and regardless of the presence of Au in the catalyst droplets. Therefore, for In–Ga droplets, we have $\beta_{g}\ll 1$ and $\beta_{l}\ll 1$, meaning that such droplets consist of almost pure In (mixed with Au in the case of Au-catalyzed VLS growth) at y→1. The same applies for vapor–solid growth under close-to-equilibrium conditions, where obtaining any appreciable fraction of In in a solid requires an In-rich vapor phase with z→1. Conversely, for Al–Ga droplets, we have $\beta_{g}\gg 1$ and $\beta_{l}\gg 1$, corresponding to a negligible amount of Al in the droplets relative to Ga at y→0, and similarly for vapor–solid growth under close-to-equilibrium conditions at z→0. While equilibrium vapor–solid distributions may not be relevant for the fast VLS growth of III–V NWs, as will be discussed shortly, the equilibrium liquid—solid distribution should describe any liquid–solid growth of VLS NWs based on group-III intermix. Both the miscibility gap region and the affinity effect cannot be fully circumvented in the equilibrium distributions.

The kinetic vapor–solid distribution given by Equation (17) is very different from the equilibrium distributions. First, this distribution is five-parametric and is reduced to the four-parametric function of Ref. [35] at large enough diffusion lengths of A and B adatoms corresponding to $\sigma_{k}R/2\lambda_{k}\to 0$ for k= A, B, which yields $c=\lambda_{A}/\lambda_{B}$. The purely kinetic parameter c describes different arrival rates of A and B adatoms into the droplet, while parameter Γ is related to the supersaturation level of vapor with respect to a solid (because it contains the ratio of the equilibrium fluxes over the actual vapor fluxes [35]). In particular, Γ is inversely proportional to $(I_{A}+I_{B})I_{C}^{1/2}$ and therefore can be decreased by increasing the total flux of group-III atoms, the flux of group-V atoms, or both fluxes. The kinetic vapor–solid distribution also contains the miscibility gap, but the ω-dependent terms in Equation (17) are proportional to Γ. Therefore, the miscibility gap can be fully circumvented by fast VLS growth kinetics at high enough vapor supersaturations corresponding to Γ→0 [24,32,33,34,35,36,37]. In simple terms, with negligible desorption fluxes, the solid composition is entirely determined by the material inputs into the droplet. At Γ→0, the vapor–solid distribution is reduced to the one-parametric Langmuir–McLean formula
(20)$$z=\frac{x}{c+\left(1-c\right)x},$$
without any thermodynamic parameters [35]. According to Equation (17) for c, the solid composition is different from the content of A atoms in vapor only due to different beam geometries in MBE or precursor decomposition efficiencies in VPE, and different diffusivities of A and B adatoms on the NW sidewalls.

Au-catalyzed Al_x_Ga_1−x_As NWs grown by MBE often feature the spontaneous formation of core–shell radial structures, where cylindrical cores have lower AlAs fractions compared to the tapered shells [21,46]. The origin of this effect is the radial vapor–solid growth of the shells around the VLS cores, where the composition of the shells is close to the vapor composition [21]. The lower AlAs fractions in the VLS cores are explained by a shorter diffusion length of Al adatoms on the NW sidewalls compared to Ga. This corresponds to c=0.385 for Al_x_Ga_1−x_As NWs grown by MBE on Si(111) substrates at 510 °C, where different vapor compositions z from 0.1 to 0.6 were obtained by varying the Al and Ga fluxes at a fixed total group-III flux yielding a 2D-equivalent AlGaAs growth rate of 0.3 nm/s with a V/III flux ratio of 3 [21,35,37]. AlGaAs is a lattice-matched system without any miscibility gaps (ω≅0). However, its equilibrium liquid–solid distribution is very asymmetric due to a large $\beta_{l}$, which equals 556 at 600 °C for self-catalyzed Al_x_Ga_1−x_As NWs [20,29]. Hence, the catalyst droplet consists of almost pure liquid Ga, with a negligible fraction of Al. No such asymmetry is present in the vapor–solid distribution of Ref. [21], where the Al content in vapor is systematically larger than the AlAs fraction in Al–GaAs NWs.

Let us now discuss the data of a highly mismatched InGaAs material system, where the miscibility gap closes for temperatures above the critical temperature of 543 °C [29]. Figure 2 shows the only published data on the liquid–solid distribution of Au-catalyzed In_x_Ga_1-x_As NWs, obtained using in situ growth and compositional monitoring of NW growth inside an environmental transmission electron microscope [15]. These InGaAs NWs were grown by Au-catalyzed MOVPE at 380 °C using 30 nm diameter colloidal Au nanoparticles, under a gas phase V/III ratio of ~1000 and variable fluxes of In and Ga precursors. The measured x(y) curve is not exactly equilibrium, as discussed in detail in Ref. [37], where a refined model with the kinetic effects included describes the data with a partially suppressed miscibility gap. However, the miscibility gap region with an almost vertical section of x(y) dependence between x≅0.1 and x≅0.9 is present. The entire curve can reasonably be fitted using the purely equilibrium shape given by Equation (5) with ω=2.724 [35] and $\beta_{l}=0.06$ (see Table 1 for a summary of the model parameters used for fitting different data).

The measured vapor–solid distributions of Au-catalyzed In_x_Ga_1−x_As NWs grown by MOVPE at 450 °C and 470 °C on InAs(111)B substrates [18], and at 420 °C on GaAs(111)B substrates [10], are very different from the liquid–solid distribution of Ref. [15], but quite similar to the vapor–solid distribution of catalyst-free selective-area In_x_Ga_1−x_As NWs grown by MOVPE at 570 °C on graphene [16]. All the measured x(z) curves shown in Figure 2 are monotonic and do not feature any effects associated with the high affinity of In with respect to Ga. The vapor–solid distributions show very similar contents of In atoms in vapor and InAs fractions in NWs. The compositional data corresponding to the tops and bottoms of Au-catalyzed NWs at 450 °C, and the average compositions of Au-catalyzed NWs NWs at 420 °C, are well-fitted by the kinetic vapor–solid distribution given by Equation (17), with the parameters summarized in Table 1. It is seen that the fitting values of Γ are quite low in all cases (Γ≤0.2). Due to the fast growth kinetics at relatively high vapor supersaturations, no miscibility gaps are present in the vapor–solid distributions of In_x_Ga_1−x_As at 420 °C and 450 °C. Overall, the measured vapor–solid distributions are centered around the simplest curve, x=z. The differences in the x(z) dependences arise due to different NW radii, surface diffusivities of In and Ga adatoms or decomposition efficiencies of In and Ga precursors (at a low temperature of 420 °C [10]), positions along the NW axis, but not by thermodynamic factors.

Kinetically driven compositional trends in InGaAs NWs are further demonstrated in Figure 3, which presents a compilation of the NW compositions at different temperatures for a similar In content in vapor between 0.27 and 0.42 [10,11,14,15,16]. Only one data point at 380 °C is outside the miscibility gap, which corresponds to close-to-equilibrium liquid–solid distribution. All the data points for Au-catalyzed InGaAs NWs grown under different conditions below the critical temperature are inside the miscibility gap, which confirms its suppression by the kinetic growth effects. Of course, this kinetic suppression of the miscibility gap is not specific for VLS NWs and has long been known for 2D epi-layers (see, for example, Ref. [47]).

Figure 4 shows the vapor–solid distributions of Au-catalyzed In_x_Ga_1−x_P NWs of Ref. [17]. These NWs were grown by MOVPE at different temperatures on InP(111)B substrates using randomly dispersed 80 nm diameter Au nanoparticles. The InGaP system is another example of a group-III-based ternary with miscibility gaps at typical growth temperatures. For growth at 460 °C and 480 °C, the ω values equal 2.514 and 2.438 [35], yielding even wider miscibility gaps than in the InGaAs system at the same temperatures. The measured vapor–solid distributions shown in Figure 4 are non-linear, which requires relatively high values of Γ=0.52 at 460 °C and Γ=0.65 at 480 °C to fit the data by Equation (17). However, no miscibility gaps are present in these distributions. For comparison, Figure 4 shows the equilibrium vapor–solid distributions of an In_x_Ga_1−x_P ternary, obtained from Equation (19) at $\beta_{g}=b_{g}=exp(\Delta \mu_{AC}^{0}-\Delta \mu_{BC}^{0})$. The values of $\beta_{g}$ are extremely small,$\;\beta_{g}=$ 0.00165 at 460 °C and 0.00196 at 480 °C, corresponding to very large In content in vapor required to obtain any appreciable fraction of InP in InGaP NWs. These equilibrium distributions also contain wide miscibility gaps. As in the case of InGaAs NWs, the real vapor–solid distributions are very far from the thermodynamically limited regime.

Highly mismatched InGaN is one of the most difficult ternary systems based on group-III intermix for obtaining high InN fractions without material segregation and compositional inhomogeneities [48,49]. The thermodynamic miscibility gap of InGaN closes at a high critical temperature of around 1250 °C [48], as shown in Figure 5. The data points show the compositions of self-catalyzed In_x_Ga_1−x_N NWs grown by HVPE on Si substrates at different temperatures from 630 °C to 680 °C, at a fixed In content in vapor of 0.6 [24]. All these data points are within the miscibility gap and are well-fitted by the simplest kinetic Langmuir–McLean formula given by Equation (20). Arrhenius-type temperature dependence, $c=const\times exp(∆E/k_{B}T)$ with ∆E= 3.29 eV, describes the enhanced desorption of In atoms from In-rich catalyst droplets at higher temperatures [24]. Therefore, the vapor–solid distribution of these InGaN NWs is also kinetically controlled, without any influence of thermodynamic factors.

## 7. Conclusions

To summarize, the developed approach for obtaining the analytical vapor–solid distribution of VLS III–V ternary NWs based on group-III intermix uses close-to-equilibrium conditions for group-III atoms at the liquid–solid interface of a growing ternary island and close-to-equilibrium conditions for highly volatile group-V atoms at a droplet surface. Whenever the liquid–solid distribution is close to equilibrium, or nucleation-limited, as given by Equation (5), and the chemical potential of group-V atoms in the droplet is close to equilibrium with vapor, as given by Equation (9), the vapor–solid distribution has the kinetic form given by Equation (17). This kinetic distribution contains no characteristics of liquid and circumvents uncertainty in the unknown group-V concentration in a catalyst droplet and other parameters of the chemical potentials of different atoms in liquid, which is an important step with respect to the previously obtained results [32,35,37]. The model fits very well the compositional data of different VLS NWs based on group-III intermix. In particular, it describes the kinetic suppression of the miscibility gaps in the vapor–solid distributions for highly mismatched InGaAs, InGaP, and InGaN NWs. We now plan to consider VLS III–V ternary NWs based on group-V intermix using a similar approach.

## Figures and Tables

**Figure 1 nanomaterials-13-02532-f001:**
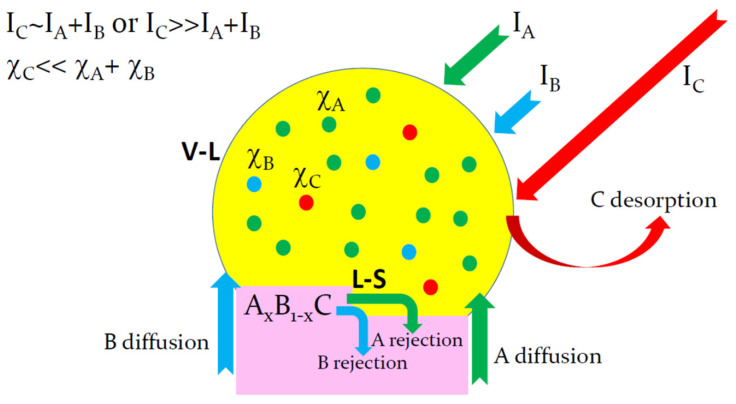
Illustration of the VLS growth process for a ternary A_x_B_1−x_C NW based on group-III intermix. The growth species are deposited from vapor fluxes I_A_, I_B_, and I_C_ such that I_C_~I_A_ + I_B_ or even I_C_ ≫ I_A_ + I_B_, which yields group-V-rich conditions at the droplet surface even in the presence of surface diffusion of group-III adatoms from the NW sidewalls into the droplet. However, the group-V concentration in the droplet is always much lower than the total group-III concentration (χ_C_ ≪ χ_A_ + χ_B_). Desorption of group-V atoms from the vapor–liquid interface V–L should therefore be very large. The liquid–solid growth of a ternary NW at the liquid–solid interface L–S is group-V-poor, which is why most group-III atoms are rejected from the island boundary. From these considerations, the limiting step for the incorporation of group-V atoms is their desorption at the V–L interface, while the limiting step for group-III atoms is their difficult incorporation at the L–S interface in the absence of group-V atoms available for the liquid–solid growth of a 2D island.

**Figure 2 nanomaterials-13-02532-f002:**
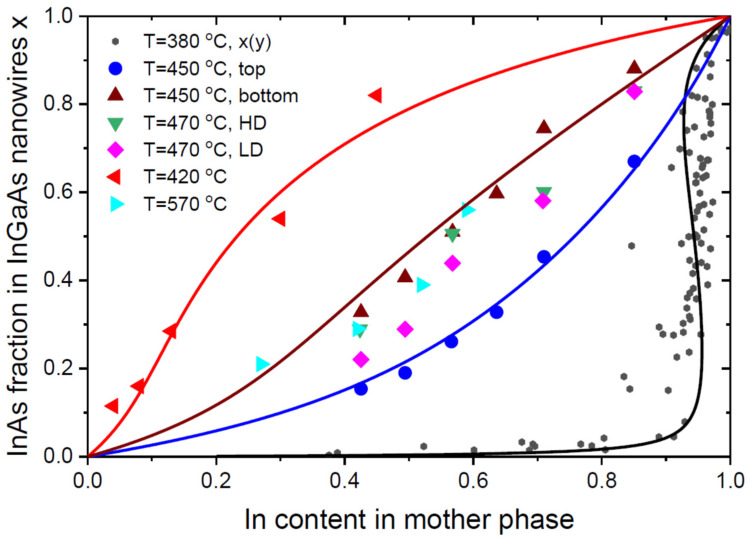
Liquid–solid distribution of Au-catalyzed In_x_Ga_1−x_As NWs grown by MOVPE in the openings of SiNx film at 380 °C [15], compared to the vapor–solid distributions of Au-catalyzed In_x_Ga_1−x_As NWs grown by MOVPE on InAs(111)B substrates at 450 °C (for which the compositions were measured at the NW tops and bottoms) and at 470 °C with a high NW surface density (HD) corresponding to an average inter-NW distance of 316 nm and a low density (LD) corresponding to an inter-NW distance of 707 nm [18]; Au-catalyzed In_x_Ga_1−x_As NWs grown by MOVPE at 420 °C on GaAs(111)B substrates [10]; and catalyst-free In_x_Ga_1−x_As NWs grown by selective-area MOVPE on graphene surface at 570 °C [16] (symbols). The liquid–solid distribution is fitted by Equation (5) and the vapor–solid distributions are fitted by Equation (17) with the parameters summarized in Table 1 (lines).

**Figure 3 nanomaterials-13-02532-f003:**
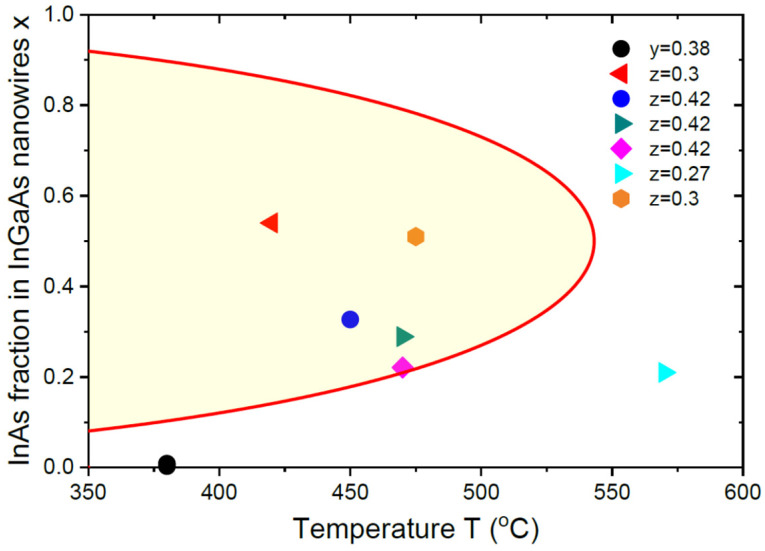
InAs fractions in InGaAs NWs at y = 0.38 [15] and z = 0.3 [10], 0.42 [11], 0.27 [16], and 0.3 [14] versus temperature (symbols). The shaded zone inside the parabola shows the miscibility gap for In_x_Ga_1−x_As system, with a critical temperature of 543 °C. All the vapor–solid data points for Au-catalyzed InGaAs NWs are inside the miscibility gap.

**Figure 4 nanomaterials-13-02532-f004:**
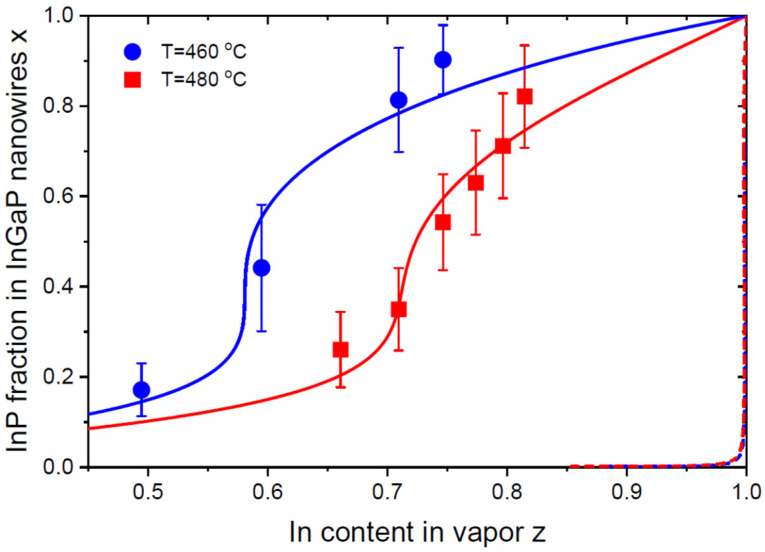
Vapor–solid distributions of Au-catalyzed In_x_Ga_1−x_P NWs grown by MOVPE at two different temperatures of 460 °C and 480 °C on InP(111)B substrates (symbols) [17], fitted by Equation (17) with the parameters given in Table 1 (solid lines). The dashed lines show the equilibrium vapor–solid distributions given by Equation (19) at *β_g_* = 0.00165 at 460 °C and 0.00196 at 480 °C. These lines are almost invisible due to a much higher affinity of In with respect to Ga in this material system, such that obtaining any appreciable amount of InP in InGaP alloy requires an almost pure In vapor.

**Figure 5 nanomaterials-13-02532-f005:**
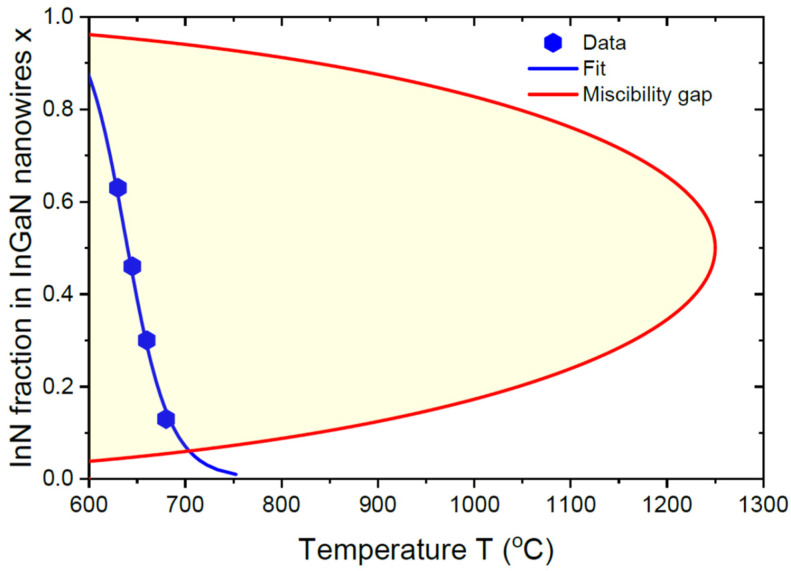
Miscibility gap of In_x_Ga_1−x_N system with a critical temperature of 1250 °C [48], shown by the shaded area inside the parabola. Symbols correspond to the measured compositions of self-catalyzed In_x_Ga_1−x_N NWs grown by HVPE on Si substrates at different temperatures from 630 °C to 680 °C, with a fixed In content in vapor z = 0.6 [24]. All these data points are inside the miscibility gap. The temperature dependence of the NW composition is fitted by the purely kinetic Langmuir–McLean formula given by Equation (20), with Arrhenius-type temperature dependence of c with an activation energy of 3.29 eV.

**Table 1 nanomaterials-13-02532-t001:** Fitting parameters for III–V NWs based on group-III intermix.

Ref.	System	T (°C)	Distribution Type	ω	$\beta_{l}$	Γ	$b_{g}$	c	$\lambda_{A}/\lambda_{B}$
[21]	Au-catalyzed AlGaAs NWs	510	x(z), kinetic	-	-	0	-	0.385	-
[15]	Au-catalyzed InGaAs NWs	380	x(y), equilibrium	2.724	0.06	-	-	-	-
[18]	Bottoms of Au-catalyzed InGaAs NWs	450	x(z), kinetic	2.373	-	0.15	0	0.39	0.39
[18]	Tops of Au-catalyzed InGaAs NWs	450	x(z), kinetic	2.373	-	0.15	0	1.2	1.2
[10]	Au-catalyzed InGaAs NWs	420	x(z), kinetic	2.475	-	0.2	0	4.1	1
[17]	Au-catalyzed InGaP NWs	460	x(z), kinetic	2.514	-	0.52	0	4.2	4.2
[17]	Au-catalyzed InGaP NWs	480	x(z),kinetic	2.438	-	0.65	0	2.2	2.2
[24]	Self-catalyzed InGaN NWs	630–680	x(z), kinetic	-	-	0	-	Arrhenius, ΔE= 3.29 eV	1

## Data Availability

Not applicable.

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
