# Peer review of "Composition of Vapor–Liquid–Solid III–V Ternary Nanowires Based on Group-III Intermix"

_nanomaterials, 2023, doi:10.3390/nano13182532_

Round 1

Reviewer 1 Report

The author develops a theory for the compositional control in III-V ternary nanowires grown by VLS. He shows that the vapor-solid distribution is kinetically controlled and the liquid-solid distribution is equilibrium or nucleation limited. The available experimental data on various ternary III-V nanowires are analyzed, and a very good agreement with the developed theory is established. The paper is clearly written, and I recommend publication in the present form.

Typos:

lines 48 and 80: replace B_{-x} with B_{1-x}

line 163: replace colume with volume

Eliminate paragraph indentation after formulas in the middle of a paragraph, like e.g. in line 114.

Author Response

The response letter is attached

Reviewer 2 Report

This paper demonstrates the theoretical modeling of VLS III-V ternary nanowires in terms of their composition AxB1-xC, where the elements A and B are of group III and the element C is of group C. The authors discuss the difference between a liquid-solid interface and a vapor-solid interface as well as kinetic suppression of miscibility gaps. They validated that previously reported experimental data of VLS grown ternary nanowires can be interpreted in the suggested model. The claimed significant impact of this study is that the suggested model circumvents the uncertainty in selecting a growth model (i.e. close-to-equilibrium or kinetic) without using unknown parameters. The study shows the advanced progress compared with their previous studies and thus worth publishing in this journal. Several concerns should be addressed prior to accept this paper.

1.       The authors describe that “the group V concentration in the droplet is in the order of 0.01 (in line 88-89)” and “A small fraction of group V atoms … will be subsequently incorporated into NW with an almost 100% probability (in line 97-99)”. Are these descriptions derived from previous reports and coincident with the reported experimental data?

2.       The former term of equation (8) must have the coefficient of 4πR^2 by assuming the liquid droplet surface area.

3.       The value of ω seem to be extracted by a data fitting but cannot be predicted. In this case, how can the model be useful for precisely tuning the composition of group III intermix?

4.       The correlation between the miscibility gap and the model seems unclear. The authors need to describe how the miscibility gap can be conducted.

5.       The inverse proportional relationship between the parameter Γ and the vapor fluxes seems to be not intuitive. Please explain why the supersaturation degree decreased while increasing vapor flux.

6.       What does it mean of “the asymmetry of equilibrium liquid-solid distribution”? The current explanation is insufficient for broad readers to imagine the phenomenon.

7.       In figure 2, I guess that y(x) must be x(y).

8.       There is typo in line 351. I guess that ‘Al’ should be ‘All’.

9.       The kinetic suppression of the miscibility gap should be more carefully explained. The current explanation is insufficient for broad readers to imagine the phenomenon.

10.   The model contains both of the equilibrium liquid-solid distribution and the kinetically controlled vapor-liquid distribution, and also covers the core-shell radial structures derived from vapor-solid growth. On the other hand, the kinetic suppression of miscibility gap seems to be discussed on the liquid-solid interface because the top of nanowire also shows the suppressed miscibility gap in figure 2. The terms of “equilibrium liquid-solid distribution” and “kinetic suppression at liquid-solid interface” are contradictive and thus confusing the readers. The authors need to provide the clear explanation.

11.   The model seems not contain the interfacial energy, which is essential in the nucleation theory and crystal growth. I guess that interfacial energy correlated with the affinity factor β. Because the equation of affinity factor seems complex, can it be simplified by using the term of interfacial energy?

Author Response

The response letter is attached
